# Laryngeal Neural Monitoring during Pediatric Thyroid Cancer Surgery—Is Transcartilage Recording a Preferable Method?

**DOI:** 10.3390/cancers13164051

**Published:** 2021-08-11

**Authors:** Tzu-Yen Huang, Hoon-Yub Kim, Gianlorenzo Dionigi, I-Cheng Lu, Pi-Ying Chang, Feng-Yu Chiang, Yi-Chu Lin, Hsin-Yi Tseng, Cheng-Hsin Liu, Che-Wei Wu

**Affiliations:** 1International Thyroid Surgery Center, Department of Otolaryngology-Head and Neck Surgery, Kaohsiung Medical University Hospital, Kaohsiung Medical University, Kaohsiung 807, Taiwan; tyhuang.ent@gmail.com (T.-Y.H.); reddust0113@yahoo.com.tw (Y.-C.L.); sycatlovestar@gmail.com (H.-Y.T.); Tim.chliu27@gmail.com (C.-H.L.); 2Department of Biological Science and Technology, National Yang Ming Chiao Tung University, Hsinchu 300, Taiwan; 3KUMC Thyroid Center, Department of Surgery, Korea University Hospital, Korea University College of Medicine, Seoul 02841, Korea; hoonyubkim@gmail.com; 4Division of Surgery, Istituto Auxologico Italiano IRCCS, 20095 Milan, Italy; gdionigi@unime.it; 5Department of Medical Biotechnology and Translational Medicine, University of Milan, 20133 Milan, Italy; 6Department of Anesthesiology, Kaohsiung Municipal Siaogang Hospital, Kaohsiung Medical University Hospital, Faculty of Medicine, College of Medicine, Kaohsiung Medical University, Kaohsiung 812, Taiwan; u9251112@gmail.com; 7Department of Anesthesiology, Kaohsiung Municipal Ta-Tung Hospital, Kaohsiung Medical University Hospital, Faculty of Medicine, College of Medicine, Kaohsiung Medical University, Kaohsiung 801, Taiwan; annabelle69@gmail.com; 8Department of Otolaryngology-Head and Neck Surgery, E-Da Hospital, Kaohsiung 824, Taiwan; fychiang@kmu.edu.tw; 9School of Medicine, College of Medicine, I-Shou University, Kaohsiung 824, Taiwan; 10Department of Otolaryngology-Head and Neck Surgery, Kaohsiung Municipal Siaogang Hospital, Faculty of Medicine, College of Medicine, Kaohsiung Medical University, Kaohsiung 812, Taiwan; 11Center for Liquid Biopsy and Cohort Research, Faculty of Medicine, College of Medicine, Kaohsiung Medical University, Kaohsiung 807, Taiwan

**Keywords:** pediatric thyroid surgery, pediatric thyroid cancer, intraoperative neuromonitoring, endotracheal tube neuromonitoring, transcartilage neuromonitoring

## Abstract

**Simple Summary:**

Comprehensive surgery is advisable in pediatric thyroid cancer. Intraoperative neuromonitoring (IONM) is a useful adjunct to thyroid surgery because it reduces recurrent laryngeal nerve (RLN) palsy risk. Use of the transcartilage (TC) recording method has recently expanded because studies in adult patients indicate that TC electrodes provide more stable electromyography (EMG) signals compared to conventional endotracheal tube (ET) electrodes. This study is the first to report the use of TC-IONM in a pediatric population. In contrast with conventional ET-IONM, TC-IONM avoids the issue of limited accessibility of ET sizes and ET malpositioning or displacement. In pediatric thyroid surgeries, the higher EMG amplitude, superior signal stability and superior signal quality in TC-IONM greatly facilitate a meticulous and extensive RLN dissection that minimizes residual thyroid tissue. In pediatric thyroid cancer, TC-IONM is feasible and effective for monitored thyroidectomy and should be considered the preferable monitoring method.

**Abstract:**

The use of transcartilage (TC) intraoperative neuromonitoring (IONM) in a pediatric population has not been reported. This study evaluated the feasibility and the benefit of using TC-IONM for thyroid cancer surgery in a pediatric population. This retrospective single-center study enrolled 33 pediatric patients who had received an IONM-assisted thyroidectomy. Demographic characteristics, standardized IONM laryngeal examinations and stimulation information (L1-V1-R1-R2-V2-L2) were compared between endotracheal tube (ET) and TC methods. In the 15 cancer patients (30 nerves), TC-IONM provided significant higher electromyography (EMG) amplitude (*p* < 0.001), signal stability (lower V1/V2 signal correlation, r = 0.955 vs. r = 0.484, *p* = 0.004), signal quality (higher ratio of V1 or V2 amplitude <500 µV, 0.0% vs. 43.8%, *p* = 0.005) and lower R1-R2p change (7.1% vs. 37.5%, *p* = 0.049) compared to ET-IONM. In the 18 benign patients (28 nerves), TC-IONM provided significantly higher EMG amplitude (*p* < 0.001), signal stability (r = 0.945 vs. r = 0.746, *p* = 0.0324) and non-significant higher signal quality and R1-R2p change. This report is the first to discuss the use of TC-IONM in pediatric thyroid surgery. In contrast with ET-IONM, TC-IONM had superior amplitude, stability and quality of EMG signals, which greatly facilitates the meticulous recurrent laryngeal nerve dissection in pediatric thyroidectomies. The TC-IONM method can be considered a feasible, effective and preferable method of monitored thyroidectomy in pediatric thyroid cancer.

## 1. Introduction

Thyroid cancer is not a common disease in pediatric populations. Pediatric thyroid cancers account for only 1.8% of all thyroid malignancies in the United States; most cases occur in the second decade of life [1]. In a Netherlands study, a pediatric population with differentiated thyroid carcinoma had an overall survival of 99.4% after a median follow-up period of 13.5 years [2]. Since pediatric thyroid cancer patients usually have a long survival time, residual tumors and recurrence of tumors are typically the major concerns, even after a complete radiological and biochemical response is achieved [3]. To prevent locoregional recurrence and to avoid further revision surgery for pediatric thyroid cancer patients, a comprehensive surgery including total thyroidectomy (TT) and lymph node management is recommended [4,5,6]. Although radically removing thyroid and lymph node tissues raises the risk of nerve injury, the risk of nerve injury is even higher in revision surgery. Additionally, potential complications of primary thyroidectomy must be carefully considered, particularly in pediatric thyroid surgery, which has a complication rate higher than that in adults [7].

Dysphonia, dysphagia and aspiration caused by recurrent laryngeal nerve (RLN) injury in pediatric patients can have negative developmental and socioemotional effects [8]. The reported rate of transient and permanent vocal cord paralysis in pediatric thyroidectomy ranges from 0 to 9.6% [5,7,9,10] and the reported rate of bilateral vocal cord paralysis in pediatric thyroidectomy is 1.5% [10]. Intraoperative neuromonitoring (IONM) of the RLN is a useful adjunct technique for gold standard RLN visualization in adult thyroid surgery and has gained widespread acceptance in the international community [11,12,13,14,15]. Routine use of IONM is recommended as it is not always possible to preoperatively predict which patients will have difficult anatomy [16]. It can also assist the clinical decision-making process involved in optimal RLN management, especially for invasive thyroid cancer surgery [15,16]. Additionally, IONM of the RLN is a commonly applied technique in pediatric patients [17,18,19,20]. Children have a thinner RLN diameter and more challenging anatomy compared to adults. A consensus statement strongly agreed that applying IONM showed benefit in the setting of bulky thyroid and lymph node disease [21].

Since the larynx and trachea are relatively small in pediatric patients, IONM techniques in pediatric patients differ from those in adults [17,22]. Endotracheal tube (ET) surface recording electrodes systems are now used worldwide for IONM-assisted thyroidectomy in adults; however, a major limitation of clinical use of ET-IONM is the need to maintain constant contact between the electrodes and vocal cords during surgery to obtain a high-quality recording. An ET that is malpositioned during intubation or displaced during surgical manipulation can cause a false decrease or false loss of electromyography (EMG) signal. False signals may be difficult to distinguish from EMG changes caused by true RLN injuries and may impair surgical decision making [23]. In pediatric thyroid surgery, the limited selection of commercially available ET-IONM sizes may result in use of an ET-IONM size that does not provide the optimal stability of contact between the ET recording electrode and the vocal cord. Confirming the ET electrode position in pediatric patients is also difficult because of the small size of the larynx.

Alternative electrode systems designed to address factors that limit ET-IONM accuracy and efficacy have been investigated recently. In experimental studies, Wu et al. [24] confirmed the hypothesis that transcartilage (TC) recording electrodes can function like the ET-based electrodes and enable access to the EMG response of the vocalis muscle originating from the thyroid cartilage inner surface. Recent clinical studies have further confirmed that, in adult thyroid surgery, TC-IONM provides significantly higher signal strength and stability and significantly less false signal compared to ET-IONM [25,26]. Therefore, TC-IONM is increasingly recognized as an acceptable alternative to ET-IONM and is gradually expanding to other aspects of thyroid surgery [27,28,29,30,31,32]. To our knowledge, however, use of this technique in pediatric populations had not been reported. Therefore, the objectives of this study were to evaluate the feasibility and benefit of using TC-IONM in pediatric thyroid cancer surgery and to determine whether TC-IONM is preferable to ET-IONM in this scenario. In order to investigate whether TC-IONM has additional benefits in pediatric patients with thyroid cancer, this study also surveys pediatric patients with benign thyroid disease for comparison.

## 2. Materials and Methods

This retrospective study enrolled pediatric patients who had received thyroid surgery from January 2011 to December 2019 at Kaohsiung Medical University Hospital. The age range of the patients was 12 to 18 years old. All patients received primary thyroid surgery performed under IONM by one of three experienced thyroid surgeons (F.-Y. C., C.-W. W and T.-Y. H) on the hospital IONM team. Standard procedures for equipment setup, anesthesia and loss of signal troubleshooting were performed by the IONM team of Kaohsiung Medical University Hospital [33,34,35,36] according to the guidelines published by the International Nerve Monitoring Study Group (INMSG) [12,13,14,15].

The ET recording electrodes were used during January 2011 until September 2015 and TC recording electrodes were used during October 2015 until December 2019. In procedures performed using ET recording electrodes, all patients were intubated with a Nerve Integrity Monitor Standard Reinforced EMG ET (Medtronic Xomed, Jacksonville, FL, USA). Two sizes of commercially available ETs are used in our institution. The ET used in all female pediatric patients and in male pediatric patients younger than 15 years has an internal diameter (ID) of 6.0 mm and an outer diameter (OD) of 8.8 mm. The dimensions of the ET used in male pediatric patients older than 15 years are ID 7.0 mm and OD 10.2 mm. Laryngofiberscopy was routinely used to verify the proper positioning of ET surface electrodes after neck positioning. In procedures performed using TC recording electrodes, paired 12 mm standard needle electrodes (Medtronic Xomed, Jacksonville, FL, USA) were gently inserted at an oblique angle into each side of the subperichondrium of the outer surface of the thyroid alar cartilage. (Figure 1)

In each thyroidectomy, RLNs were routinely identified and preserved by visualization with the adjunct of IONM to minimize perioperative complications. Standard IONM procedures were strictly followed in all patients regardless of electrode type and EMG signals with the largest amplitude (V1-R1-R2-V2) were recorded [12,33,34]. After thyroidectomy, the exposed RLN was routinely tested at the lowest proximal end (R2p signal) and at the most distal end near the laryngeal entry point (R2d signal) to determine whether there is RLN injury site in the exposed and dissected segment [35]. In all patients, the external branch of the superior laryngeal nerve (EBSLN) function was evaluated by observing the cricothyroid muscle (CTM) twitch induced by stimulation of the most proximal end of the EBSLN after dissection of the superior thyroid pole [13,37].

Laryngofiberscopic examination for vocal cord mobility before (L1) and after (L2) surgery was video recorded in all patients. Serum ionized calcium (iCa) levels were measured in all patients undergoing total thyroidectomy before and 12, 24, 48 and 72 h after surgery. Postoperative hypocalcemia was defined as iCa under 4.2 mg/dL in at least two measurements. Hypocalcemia was considered permanent if a patient with hypocalcemia still required calcium supplements 12 months after surgery.

To analyze the variables, independent t test, Pearson chi-square test and correlation coefficient were performed using SPSS (Version 18.0 for windows; SPSS Inc., Chicago, IL, USA). A two-tailed *p* value less than 0.05 was considered statistically significant.

## 3. Results

### 3.1. Subject Characteristics

Table 1 presents the demographic and clinical characteristics of the 15 pediatric patients with thyroid cancer (five males and ten females, mean age 17.3 years) and the 18 pediatric patients with benign thyroid disease (three males and fifteen females, mean age 17.2 years). Among the 30 at-risk RLNs in thyroid cancer patients, 16 (53.3%) nerves were monitored by ET-IONM and 14 (46.7%) nerves were monitored by TC-IONM. Among 28 at-risk RLNs in benign thyroid disease patients, 11 (39.3%) nerves were monitored by ET-IONM and 17 (60.7%) nerves were monitored by TC-IONM. The pre and postoperative laryngeal examination (L1 and L2) showed that no patients had asymmetric vocal cord movement. All patients with thyroid cancer received TT. In patients with benign thyroid disease, ten (55.6%) received TT, including five (27.8%) patients with thyroiditis or Graves’ disease.

The pathology reports for the patients with thyroid cancer revealed thirteen (86.7%) patients with papillary thyroid carcinoma (ten classic, one follicular variant, one cribriform-morular variant, and one diffuse sclerosing variant) and two (13.3%) patients with follicular thyroid carcinoma. The average tumor volume was 9.0 cm^3^ (range, 0.1 to 32.8 cm^3^). In the nine patients who had papillary thyroid carcinoma and documented BRAF^V600E^ results, three had positive results and six had negative results. The two patients with follicular thyroid carcinoma both had negative RAS results. The T stage distribution was one, four, four, one and five patients with stages T1a, T1b, T2, T3a and T3b, respectively. The distribution of N stage was seven, four and four patients with stages N0, N1a and N1b, respectively. Regarding radioactive iodine (RAI) therapy, three patients did not receive RAI therapy, while one, three, six and two patients received RAI therapy with 50, 100, 150 and ≥200 mCi doses, respectively. Four (26.7%) cancer patients and two (20.0%) benign patients suffered transient postoperative hypocalcemia. None of the pediatric patients had permanent postoperative hypocalcemia or postoperative hematoma.

### 3.2. Stability and Quality of IONM Signals in Pediatric Surgeries

Figure 2 is a scatter plot of V1 and V2 EMG signals according to patient characteristics. The EMG signal stability was defined as the correlation coefficient (r) between the V1 and V2 signals, and a coefficient closer to 1 is considered preferable. The EMG signal quality was defined as the ratio of nerves with V1 or V2 amplitude <500 µV, and a lower ratio is considered preferable. In the malignant group, TC-IONM provided significantly higher signal stability (*p* = 0.0004) and quality (0.0% vs. 43.8%, *p* = 0.005) compared to ET-IONM. In the benign group, TC-IONM provided significantly higher signal stability (*p* = 0.0324) and non-significant higher signal quality (0.0% vs. 18.2%, *p* = 0.068) compared to ET-IONM. Use of ET-IONM method, malignant patients had non-significantly lower signal stability (lower V1/V2 signal correlation, *p* = 0.1660) and non-significantly lower signal quality (higher ratio with V1 or V2 <500 µV, 43.8% vs. 18.2%, *p* = 0.231) compared to benign patients. Additionally, use of TC-IONM method, no benign or malignant patients had V1 or V2 amplitude <500 µV.

Table 2 presents the IONM information for the benign and malignant groups. In both benign and malignant groups, TC-IONM obtained significantly (*p* < 0.001) higher EMG signal amplitudes (V1-R1-R2p-R2d-V2) compared to ET-IONM. No patients in this study had a R2p-R2d change >20%, which indicates that no patients had an injury to the dissected RLN segment. In the malignant group, TC-IONM provided significantly lower R1-R2p change >20% rate (7.1% vs. 37.5%, *p* = 0.049) compared to ET-IONM. In the benign group, the R1-R2p change >20% rate showed no significant difference between TC-IONM and ET-IONM (5.9% vs. 18.2%, *p* = 0.304). The false EMG signal changes caused by the potential vicissitudes of ET electrode position occur more frequently in malignant surgeries than in benign surgeries. No patients had impaired CTM twitch.

## 4. Discussion

In this study, we evaluated the feasibility and benefit of the TC-IONM method (Figure 1) for pediatric thyroid surgery in comparison with conventional ET-IONM. Standardized IONM laryngeal examinations and stimulation information (L1-V1-R1-R2-V2-L2) were successfully recorded and compared between the ET and TC methods. No patients suffered postoperative vocal cord palsy, permanent hypocalcemia or postoperative hematoma (Table 1). The IONM information confirmed that no patients had RLN injury on the dissected segment and that all patients had intact CTM twitch after EBSLN stimulation. Compared to the ET-IONM method, the TC-IONM method obtained higher EMG amplitude, better signal stability and quality in both the malignant group and the benign group. Notably, pediatric cancer surgeries performed using ET-IONM have a higher rate of false EMG signal change by the potential vicissitudes of ET electrode position and the overall EMG signal showed less stability and lower quality than in benign surgeries (Table 2 and Figure 2).

An RLN injury resulting in vocal cord paralysis remains a major source of morbidity after thyroid surgeries. Vocal cord paralysis is particularly distressing in pediatric patients because of its potential negative developmental and socioemotional effects [8]. In both adult and pediatric thyroid surgeries, however, IONM has gained increasing acceptance as an adjunct to the standard practice of visual RLN identification. Although the EMG ET recording electrode is now widely used in IONM, its major limitation is the need to maintain constant contact between the electrode and vocal cords during surgery to obtain a high-quality recording [22,38,39,40]. A tube that is malpositioned during intubation (e.g., due to rotation or due to incorrect tube size or insertion depth) or displaced during neck extension or surgical manipulation (e.g., patient positioning from intubation to neck extension, swallowing or other movement of the patient caused by a light plane of anesthesia, or surgery-related traction or pressure to the laryngo-trachea and anesthesia circuit, all of which can cause the tube to rotate or migrate from the optimal placement) can cause a decrease or complete loss of EMG signal upon neural stimulation [22,32,33,38,39,40,41]. A false positive decrease or loss of signal may result in inappropriate surgical decisions, and verification or readjustment of the ET position can be complicated and time-consuming. Therefore, alternative methods such as TC-IONM have been studied in recent years to minimize factors that negatively affect ET-IONM accuracy and efficacy [23,24,25,26,27,28,29,30,31].

Since pediatric thyroid cancer surgery requires more meticulous and extensive RLN dissection to reduce residual thyroid tissue, more displacement and rotation would be applied on the soft and small caliber trachea and its adjacent structures (i.e., RLN) during lateral and medial thyroid retraction. Therefore, applicability of a neuromonitoring recording method in pediatric thyroid surgery depends on the stability and quality (>500 µV) of the IONM EMG signal obtained by the method. Compared to the ET-IONM method, the TC-IONM method used in this study obtained EMG signals with higher amplitude, stability and quality. The superior signal monitoring performance of TC-IONM enhanced surgical precision and efficiency in all steps of thyroid dissection in which RLN stress or injury may occur.

Another drawback of using the ET-IONM method in pediatric thyroid surgery is its poor accessibility. Although ET-IONM is the most popular recording method worldwide, commercially available tube sizes appropriate for pediatric patients are limited. For example, the standard Medtronic Xomed ET-IONM has a minimum ID of 6.0 mm and the Trivantage ET has a minimum ID of 5.0 mm (OD, 6.5 mm). Before surgery, the anesthesiologist should confirm that the selected ET has an OD that is appropriate for the patient. The anesthesiologist should also check for leaks before cuff inflation to ensure that the tube has a good fit and the ET position should be verified after the final positioning of the patient for surgery. For a pediatric intubation, tubes of various sizes should be available for replacement at any time during surgery, but it shows difficulty in execution when using a commercialized EMG tube. Several alternative electrode types that can be used for ET-IONM have been described in previous studies. (1) ET with adhesive surface electrodes. To monitor the vocalis muscle, adhesive electrodes can be manually attached to the tube cuff of an ET of any size [18,42,43]. (2) Postcricoid electrodes. After intubation of a pediatric patient, extraluminal electrode pads are placed in the hypopharynx under direct visualization to record EMG signals from the posterior cricoarytenoid muscle [17,44,45]. (3) Endolaryngeal hookwire electrodes. Under laryngoscopy, this electrode type is placed directly into the vocalis muscle after intubation [46,47]. Each of these techniques has associated risks when performing cuff-free tracheal intubation in patients younger than 8 years. All three intra-laryngopharyngeal methods require confirmation or manipulation of electrode placement under laryngoscopy. The EMG signal quality and stability can be degraded by laryngeal mucosal edema caused by repeated stimulation and the remaining swallowing reflex.

As in previous works that have reported on the use of the TC-IONM method in adults, we used commercially available needle electrodes. These were inserted into the subperichondrium of the outer surface of the thyroid cartilage without penetrating the inner aspect of the thyroid cartilage or intrinsic laryngeal muscles (Figure 1). No procedure-related complications occurred, and the EMG signals collected by TC-IONM had stable amplitudes and high signal quality. Unlike the adult TC-IONM method which is prone to fail to penetrate the needle or receives a reduced EMG signals block by the calcifications in elders, the pediatric patients rarely had the thyroid cartilage calcification issue and setup time was around 1 minute in all cases. Recently, Bois et al. [48] proposed a novel method for pediatric thyroid surgery using the same paired needle electrodes our study used. The needle was placed through the cricothyroid membrane and oriented in the cranial direction and kept submucosal just below the level of the vocal folds. Although this method may achieve higher EMG amplitudes, it is invasive and presents a risk of CTM injury, laryngeal submucosa hematoma and it cannot completely avoid laryngeal mucosa penetration. As an alternative option of TC-IONM without elevating the skin on the thyroid cartilage for minimally invasive surgery or even remote surgery, in these scenarios, percutaneous TC-IONM using long needle electrodes showed great potential [29,31]. Remote thyroidectomy includes endoscopic [49] and robotic [50] approaches. These had been reported as emerging surgical procedures in the pediatric population. The demand for research about full percutaneous TC-IONM in pediatric remote thyroidectomy would be much higher than that in the adult population.

Compared with other alternative recording methods used in pediatric thyroid surgery, TC-IONM has three advantages. First, TC-IONM is simple. Since needle electrodes are placed on the perichondrium of thyroid cartilage, laryngoscopy is not needed to confirm the position of the electrode and to ensure that the mucosa is not penetrated. Second, TC-IONM is cost effective. Needle electrodes used in TC-IONM are less expensive than those used in ET-IONM and other methods. TC-IONM also avoids the need to change an improper size of ET-IONM and is more affordable for patients. Third, TC-IONM is safer and less invasive than other methods but still provides EMG signals with comparable stability and quality. Information provided by TC-IONM increases the safety of surgery by providing surgeons with essential information in real time. Notably, continuous IONM (C-IONM) used in real-time monitoring is particularly sensitive to mechanical injury in adult thyroid surgery. Schneider et al. [16] applied C-IONM in children undergoing thyroidectomy procedures and concluded that C-IONM was the preferred monitoring method given the high-quality signals generated in children aged 6 years and older. In our experience, integrating the TC recording method in C-IONM can further improve signal stability in the adult population. According to the findings in this study, TC recording will be a better choice than ET recording when performing C-IONM in pediatric thyroidectomy. However, we believe that development of a modified continuous stimulation system with reduced vagus nerve dissection (i.e., patch stimulator design) could be a future research direction for a safer C-IONM in pediatric thyroidectomy.

Some limitations of this study are noted. First, this study had a small number of enrolled patients. Second, this was not a prospective randomized study, and the ET and TC procedures were not performed in the same period. Third, since no patients in this study were younger than 11 years, the practicability of TC-IONM in younger patients needs further verification. Last, in some patients, a significant decrease in the EMG signal caused by ET rotation or malposition during ET-IONM required a readjustment of the ET position to ensure that the surgery and neural monitoring could be performed safely. Therefore, the actual signal stability and signal quality in ET-IONM procedures were worse than the recorded data indicated.

## 5. Conclusions

This study is the first to report the use of TC-IONM in a pediatric population. In contrast with conventional ET-IONM, TC-IONM is not limited by the ET size or by ET accessibility. Additionally, TC-IONM is unaffected by ET malpositioning or displacement. In pediatric thyroid surgeries, the higher EMG amplitude, superior signal stability and superior signal quality in TC-IONM greatly facilitate a meticulous and extensive RLN dissection that minimizes residual thyroid tissue. Thus, in pediatric thyroid cancer, TC-IONM is feasible and effective for monitored thyroidectomy and should be considered the preferable monitoring method.

## Figures and Tables

**Figure 1 cancers-13-04051-f001:**
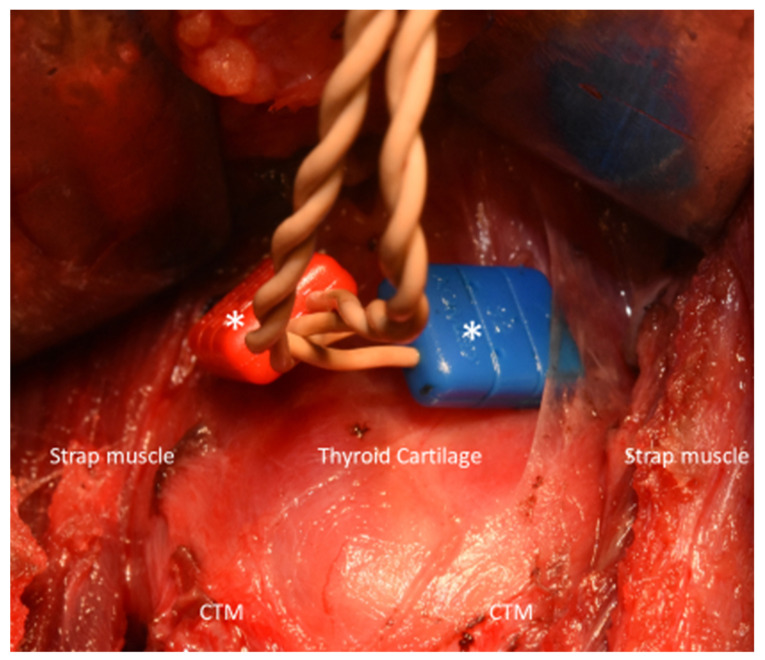
Transcartilage intraoperative neuromonitoring (TC-IONM) performed in a 17-year-old female who had received total thyroidectomy and bilateral central neck dissection of a papillary thyroid carcinoma. Paired 12 mm standard needle electrodes (white asterisk, Medtronic Xomed, Jacksonville, FL, USA) were inserted obliquely into each side of the perichondrium of thyroid cartilage. CTM= cricothyroid muscle.

**Figure 2 cancers-13-04051-f002:**
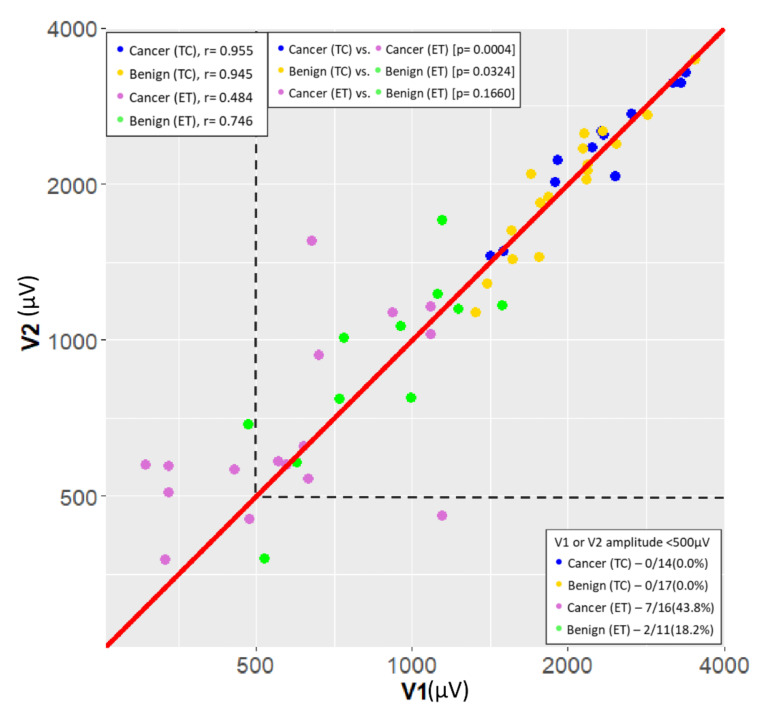
Scatterplot (logarithmic scale) of the correlation between V1 signal (µV) and V2 signal (µV), and the correlation coefficient (r) in the four groups. Blue point = Cancer (TC) group: thyroid cancer patient with surgery performed under TC-IONM, r = 0.955; Yellow point = Benign (TC) group: benign thyroid disease patient with surgery performed under TC-IONM, r = 0.945; Purple point = Cancer (ET) group: thyroid cancer patient with surgery performed under ET-IONM, r = 0.484; Green point = Benign (ET) group: benign thyroid disease patient with surgery performed under ET-IONM, r = 0.746. Cancer (TC) group showed significantly higher signal stability compared to the Cancer (ET) group, *p* = 0.0004. Benign (TC) group showed significantly higher signal stability compared to the Benign (ET) group, *p* = 0.0324. Cancer (ET) group showed non-significantly lower signal stability compared to the Benign (ET) group, (*p* = 0.0160). However, 7/16 nerves (43.8%) in Cancer (ET) group with V1 or V2 amplitude <500 µV showed higher rate than 2/11 nerves (18.2%) in Benign (ET) group. Additionally, no nerves in the Cancer (TC) group or the Benign (TC) group had V1 or V2 amplitude <500 µV. Overall, TC-IONM had higher signal stability and quality compared to ET-IONM in the thyroid cancer patients. TC = transcartilage; ET = endotracheal tube; IONM = intraoperative neuromonitoring.

**Table 1 cancers-13-04051-t001:** Demographic and clinical characteristics of pediatric patients received monitored thyroidectomy.

	Cancer	Benign
Total, n	15 patients (30 nerves)	18 patients (28 nerves)
Male, n (%)/Female, n (%)	5 (33.3)/10 (66.7)	3 (16.7)/15 (83.3)
Age, y (mean ± SD)	17.3 ± 1.6	17.2 ± 1.5
IONM recording Method	ET = 16 (53.3%)	ET = 11 (39.3%)
	TC = 14 (46.7%)	TC = 17 (60.7%)
Laryngofiberscopy		
Pre-(L1)/Post-(L2)operative VCP	0/0	0/0
Surgical extent	TT + CND = 11 (73.3%)	TL = 8 (44.4%)
	TT + CND + Unilateral LND = 3 (20.0%)	TT = 10 (55.6%)
	TT + CND + Bilateral LND = 1 (6.7%)	
Pathology report	PTC = 13 (86.7%)	Nodule or adenoma = 13 (72.2%)
	FTC = 2 (13.3%)	Thyroiditis or Graves’ = 5 (27.8%)
Surgical complications		
Transient postoperative hypocalcemia	4 (26.7%)	2 (20.0%) of 10 TT
Permanent postoperative hypocalcemia	0	0
Postoperative hematoma	0	0

Abbreviations: SD = standard deviation; IONM = intraoperative neuromonitoring; ET = endotracheal tube; TC= transcartilage; VCP = vocal cord paralysis; TT = total thyroidectomy; CND = central neck dissection; LND = lateral neck dissection; PTC = papillary thyroid carcinoma; FTC = follicular thyroid carcinoma.

**Table 2 cancers-13-04051-t002:** Comparison between intraoperative neuromonitoring information with different recording methods in pediatric patients with malignant and benign thyroid disease.

	Malignant			Benign		
Number of Patients (Nerve at Risk)	15 (30NAR)		*p* Value	18 (28NAR)		*p* Value
Recording Electrode	ET	TC	ET vs. TC	ET	TC	ET vc TC
RLN at risk (nerves)	16	14		11	17	
EMG amplitude (µV)						
V1 (mean ± SD)	634 ± 280	2328 ± 608	<0.001	905 ± 324	2048 ± 552	<0.001
R1 (mean ± SD)	1059 ± 605	3199 ± 926	<0.001	1188 ± 494	2537 ± 650	<0.001
R2p (mean ± SD)	1051 ± 489	3286 ± 850	<0.001	1269 ± 561	2662 ± 617	<0.001
R2d (mean ± SD)	1054 ± 489	3331 ± 847	<0.001	1327 ± 589	2685 ± 641	<0.001
V2 (mean ± SD)	728 ± 331	2404 ± 562	<0.001	956 ± 367	2066 ± 592	<0.001
R2p-R2d change >20%	0/16 nerves	0/14 nerves		0/11 nerves	0/17 nerves	
R1-R2p change >20%	6/16 nerves (37.5%)	1/14 nerves (7.1%)	0.049	2/11 nerves (18.2%)	1/17 nerves (5.9%)	0.304
Impaired CTM twitch	0/16 sides	0/14 sides		0/11 sides	0/17 sides	

Abbreviations: ET = endotracheal tube; TC = transcartilage; SD = standard deviation; CTM = cricothyroid muscle.

## Data Availability

The original contributions presented in the study are included in the article. Further inquiries can be directed to the corresponding authors.

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
