# Peer review of "Laryngeal Neural Monitoring during Pediatric Thyroid Cancer Surgery—Is Transcartilage Recording a Preferable Method?"

_cancers, 2021, doi:10.3390/cancers13164051_

Round 1
Reviewer 1 Report
Huang et al. reported a retrospective study evaluating the feasibility and the benefit of using transcartilage (TC) intraoperative neuromonitoring (IONM) for thyroid cancer surgery in a pediatric population.
The study is innovative and conclusions are both clear and adequately supported by results.
Few comments are suggested to improve on the submitted manuscript:
- Please, report a more exhaustive description of procedures to minimize perioperative complications in surgery for thyroid cancer. Please, see and cite: PMID: 29546741
- The design of this study should be better described. In addition, the title is focused on “thyroid cancer surgery”, while the study enrolled 18 patients with thyroid benign diseases.
- Page 9: another limitation is the small number of enrolled patients.
- The authors should better develop the future perspectives of this study.
- Tables 1 and 2. Please, indicate the type of measure adopted in the table. Mean +/- SD, mean +/- SED, median ……??!!
Author Response
Author's Response
Dear Reviewer,
We deeply appreciate your comments.
We have revised our manuscript in-line with the comment made.
The followings are our response:
Response to the Reviewer #1
Huang et al. reported a retrospective study evaluating the feasibility and the benefit of using transcartilage (TC) intraoperative neuromonitoring (IONM) for thyroid cancer surgery in a pediatric population.
The study is innovative and conclusions are both clear and adequately supported by results.
Few comments are suggested to improve on the submitted manuscript:
Comment-1:
Please, report a more exhaustive description of procedures to minimize perioperative complications in surgery for thyroid cancer. Please, see and cite: PMID: 29546741
Response:
Thank you for your suggestion. We added a description in Introduction as “Routine use of IONM is recommended as it is not always possible to predict preoperatively which patients will have difficult anatomy [16]. It can also assist the clinical decision-making process involved in optimal RLN management, especially for invasive thyroid cancer surgery [15,16].” The article was also cited as your suggestion.
Comment-2:
The design of this study should be better described. In addition, the title is focused on “thyroid cancer surgery”, while the study enrolled 18 patients with thyroid benign diseases.
Response:
Thank you for the suggestion. We added a description for design of this study in Introduction as “In order to investigate whether TC-IONM has additional benefits in pediatric patients with thyroid cancer, this study also surveys pediatric patients with benign thyroid dis-ease for comparison.”
Comment-3:
Page 9: another limitation is the small number of enrolled patients.
Response:
We totally agreed that and added a description in Discussion/Limitation as “First, this study had a small number of enrolled patients.” Thank you for the suggestion.
Comment-4:
The authors should better develop the future perspectives of this study.
Response:
Thank you for the suggestion. We added the paragraphs in Discussion as “Remote thyroidectomy includes endoscopic [49] and robotic [50] approach had been re-ported as an emerging surgical procedures in pediatric population, the demand for research about full percutaneous TC-IONM in pediatric remote thyroidectomy would be much higher than that in adult.”, and “However, we believe that development of a modified continuous stimulation system with reduced vagus nerve dissection (i.e. patch stimulator design) could be a future re-search direction for a safer C-IONM in pediatric thyroidectomy.”
Comment-5:
Tables 1 and 2. Please, indicate the type of measure adopted in the table. Mean +/- SD, mean +/- SED, median ……??!!
Response:
Thank you for the suggestion. We have modified the descriptions (mean±SD) in Table 1 and 2 as your suggestion.
We thank you for your valued comments and suggestions, which we feel substantially improve our manuscript, and hope that that the revisions meet with your approval.
Sincerely,
Tzu-Yen Huang, M.D. and Che-Wei Wu, M.D., Ph.D. (On behalf of all coauthors)
July 22, 2021

Reviewer 2 Report
Recurrent laryngeal nerve (RLN) injury resulting in transient or permanent vocal cord paresis is a main surgical complication after thyroid surgery, which significantly decreases patients’ quality of life.
Preservation of a proper function of RLN is particularly important in the pediatric population. Intraoperative Neuromonitoring (IONM) is a widely accepted procedure to improve RLNs and the external branch of the superior laryngeal nerve (EBSLN) function safety.
This is a well written, very interesting study (also from the linguistic perspective, but maybe in this sentence „The average tumor size was 9.0 cm3 (range, 0.1 to 32.8 cm3) is better to use "volume") presenting the advantages of transcartilage (TC) IONM in the pediatric population. The Authors performed a retrospective comparison between TC-IONM and conventional endotracheal tube (ET) IONM in 33 children, aged 11-18 who underwent surgery due to malignant and benign thyroid disease and proved that TC-IONM method obtained higher EMG amplitude, better signal stability and the quality in both the malignant group and the benign group.
They also presented limitations of the study in a very good manner e.g. retrospective analysis of a small population for whom ET and TC procedures were performed in a different period. The Authors noted that each child was more than 11 years of age and the practicability of TC-IONM in younger patients needs to be verified.
Apart from the above, it is the first study about the use of TC-IONM in the pediatric population describing the advantages of the recording methods and I am of the opinion that it should be published.
I only wonder if the Authors currently use continuous IONM (C-IONM), recommended in children in some European Centers (for instance in Germany and in Poland). In the Discussion section, they mentioned: ”In our experience, integrating TC recording method in C-IONM can further improve signal stability”. I am highly interested in the results of such a study.
Author Response
Author's Response
Dear Reviewer,
We deeply appreciate your comments.
We have revised our manuscript in-line with the comment made.
The followings are our response:
Response to the Reviewer #2
Recurrent laryngeal nerve (RLN) injury resulting in transient or permanent vocal cord paresis is a main surgical complication after thyroid surgery, which significantly decreases patients’ quality of life.
Preservation of a proper function of RLN is particularly important in the pediatric population. Intraoperative Neuromonitoring (IONM) is a widely accepted procedure to improve RLNs and the external branch of the superior laryngeal nerve (EBSLN) function safety.
Comments-1:
This is a well written, very interesting study (also from the linguistic perspective, but maybe in this sentence. The average tumor size was 9.0 cm3 (range, 0.1 to 32.8 cm3) is better to use "volume") presenting the advantages of transcartilage (TC) IONM in the pediatric population.
Response:
Thank you for the suggestion. We have modified the “size” to “volume” as your suggestion.
Comments-2:
The Authors performed a retrospective comparison between TC-IONM and conventional endotracheal tube (ET) IONM in 33 children, aged 11-18 who underwent surgery due to malignant and benign thyroid disease and proved that TC-IONM method obtained higher EMG amplitude, better signal stability and the quality in both the malignant group and the benign group.
They also presented limitations of the study in a very good manner e.g. retrospective analysis of a small population for whom ET and TC procedures were performed in a different period. The Authors noted that each child was more than 11 years of age and the practicability of TC-IONM in younger patients needs to be verified.
Apart from the above, it is the first study about the use of TC-IONM in the pediatric population describing the advantages of the recording methods and I am of the opinion that it should be published.
I only wonder if the Authors currently use continuous IONM (C-IONM), recommended in children in some European Centers (for instance in Germany and in Poland). In the Discussion section, they mentioned: ”In our experience, integrating TC recording method in C-IONM can further improve signal stability”. I am highly interested in the results of such a study.
Response:
Thank you very much for your review and your encouraging comments on this article. To clarify this issue, we added a paragraph in Discussion as “In our experience, integrating TC recording method in C-IONM can further improve signal stability in adult population. According to the findings in this study, TC recording will be a better choice than ET recording when performing C-IONM in pediatric thyroidectomy. However, we believe that development of a modified continuous stimulation system with reduced vagus nerve dissection (i.e. patch stimulator design) could be a future re-search direction for a safer C-IONM in pediatric thyroidectomy.”
We thank you for your valued comments and suggestions, which we feel substantially improve our manuscript, and hope that that the revisions meet with your approval.
Sincerely,
Tzu-Yen Huang, M.D. and Che-Wei Wu, M.D., Ph.D. (On behalf of all coauthors)
July 22, 2021
